# Machine Learning Approach for the Prediction of In-Hospital Mortality in Traumatic Brain Injury Using Bio-Clinical Markers at Presentation to the Emergency Department

**DOI:** 10.3390/diagnostics13152605

**Published:** 2023-08-05

**Authors:** Ahammed Mekkodathil, Ayman El-Menyar, Mashhood Naduvilekandy, Sandro Rizoli, Hassan Al-Thani

**Affiliations:** 1Clinical Research, Trauma and Vascular Surgery, Hamad Medical Corporation, Doha P.O. Box 3050, Qatar; mekkodathil@yahoo.co.uk; 2Clinical Medicine, Weill Cornell Medical College, Doha P.O. Box 24144, Qatar; 3Data Science, Alpineaid Management, Ernakulam 682304, India; mashhoodnk@gmail.com; 4Trauma Surgery Section, Hamad General Hospital (HGH), Doha P.O. Box 3050, Qatar; srizoli@hamad.qa (S.R.);

**Keywords:** trauma, brain injury, head, machine learning, predictors, support vector machine

## Abstract

Background: Accurate prediction of in-hospital mortality is essential for better management of patients with traumatic brain injury (TBI). Machine learning (ML) algorithms have been shown to be effective in predicting clinical outcomes. This study aimed to identify predictors of in-hospital mortality in TBI patients using ML algorithms. Materials and Method: A retrospective study was performed using data from both the trauma registry and electronic medical records among TBI patients admitted to the Hamad Trauma Center in Qatar between June 2016 and May 2021. Thirteen features were selected for four ML models including a Support Vector Machine (SVM), Logistic Regression (LR), Random Forest (RF), and Extreme Gradient Boosting (XgBoost), to predict the in-hospital mortality. Results: A dataset of 922 patients was analyzed, of which 78% survived and 22% died. The AUC scores for SVM, LR, XgBoost, and RF models were 0.86, 0.84, 0.85, and 0.86, respectively. XgBoost and RF had good AUC scores but exhibited significant differences in log loss between the training and testing sets (% difference in logloss of 79.5 and 41.8, respectively), indicating overfitting compared to the other models. The feature importance trend across all models indicates that aPTT, INR, ISS, prothrombin time, and lactic acid are the most important features in prediction. Magnesium also displayed significant importance in the prediction of mortality among serum electrolytes. Conclusions: SVM was found to be the best-performing ML model in predicting the mortality of TBI patients. It had the highest AUC score and did not show overfitting, making it a more reliable model compared to LR, XgBoost, and RF.

## 1. Introduction

Approximately 70 million people worldwide are victims of traumatic brain injury (TBI), which has a devastating impact on their lives. Among all forms of injuries, TBI is linked to high rates of mortality and permanent disability, with an estimated 1.5 million fatalities and several million emergency procedures each year [1,2,3,4,5]. This trend is evident across the Middle East, where the average rate of TBI per 100,000 people is 45, with a fatality rate of 10% for those treated in emergency rooms and a 25% mortality rate for those treated in intensive care units (ICU) [6]. Accurate prediction of the risk of death following a TBI in the early stages of treatment is of paramount importance, as it enables clinical decisions to be made with precision and healthcare resources to be allocated with efficiency.

Several studies have employed various Machine Learning (ML) and data mining techniques to predict the outcomes of patients including TBI [7,8,9,10,11,12,13,14,15,16,17,18,19,20,21,22,23]. The ML models have exhibited promise in predicting mortality among hospitalized patients with TBI by analyzing extensive data, including demographic information, medical history, and physiological data. These studies focused on different target populations, including adults [9,22,23,24,25] and pediatric patients [25,26,27], as well as different mechanisms of injury and severity levels of TBI [9,25,28,29,30,31,32]. The sample sizes of these studies varied significantly, ranging from hundreds [22,24,26,27,28,31] to several thousand [22,23,25,30]. The outcomes and complications predicted by machine learning approaches have also shown variations. These include the Glasgow Outcome Score (GOS) [26,32], the Glasgow Coma Score (GCS) at discharge [33], Glasgow Outcome Scale Extended (GOS-E) [19,20,31], intracranial pressure [34,35], the Barthel Index [36], the Functional Independence Measure (FIM) [24], in-hospital electrolyte imbalance [37], TBI-induced coagulopathy [38], ventilator-associated pneumonia [39], hospital length of stay [40], in-hospital mortality [9,18,19], and 30-day mortality [21]. A wide range of machine learning techniques have been employed in these studies, including Logistic Regression (LR), Support Vector Machines (SVMs), Decision Trees (DTs), Naive Bayes (NB), Artificial Neural Networks (ANNs), k-Nearest Neighbors (k-NNs), Random Forest (RF), Extreme Gradient Boosting (XGBoost), Deep Learning (DL), Gradient Boosting Trees (GBTs), Linear Discriminant Analysis (LDA), Partial Least Squares Regression (PLSR), and Convolutional Neural Networks (CNNs). Additionally, combinations of these techniques have been used, such as RF and SVM, NB, RF, k-NN, and RI [18,19,20,21,22,23,24,25,26,27,28,29,30,31,32,33,34,35,36,37,38,39,40]. Previously, it was shown that the effectiveness of SVM, RF, and NB in predicting mortality was between 67% and 97% [9,10,11].

Given that the use of ML techniques in trauma is still in the emerging phase, it remains inconclusive in the literature regarding the most appropriate method to be used and which performance indicators are most suitable for validation. The choice of method and performance indicators depends on the study’s objectives, the outcomes being measured, and the sample size. The performance indicators utilized in previous studies included Accuracy, Precision, F1 score, Area Under the Receiver Operating Characteristic Curve (AUROC), Area Under the Precision–Recall Curve (AUPRC), Sensitivity, Specificity, F-Score, Negative Predictive Value (NPV), Positive Predictive Value (PPV), and others [18,19,20,21,22,23,24,25,26,27,28,29,30,31,32,33,34,35,36,37,38,39,40].

The present study aims to explore the application of ML techniques as a mortality prediction model specifically for TBI patients in Qatar who are hospitalized, including those in the intensive care units (ICUs). The focus is on utilizing easily available bio-clinical parameters at the time of admission to enable timely interventions and enhance favorable patient outcomes. The study population represents a nationally representative sample as it is based on the national trauma registry in the country. Being a rapidly developing country in the Arabian Gulf region with a small geographical size and population, limited research has been conducted in this specific region regarding the utilization of ML techniques, particularly in the context of TBI patients. By conducting this study in Qatar, valuable insights can be gained that are specific to the region, contributing to the existing body of knowledge in the field of ML applied to TBI. The findings of this study have the potential to enhance medical practices and inform decision-making processes in the local healthcare system, ultimately improving patient outcomes in Qatar and potentially serving as a reference for similar settings in the region. The majority of TBI victims in Qatar are young males, and the utility of ML for early triage and prediction would be useful to save more lives. In clinical practice, the data obtained during the initial hours of admission can be inputted into the trained ML model to generate a predictive response. This outcome can then be used to alert clinicians about the patient’s condition and validate the model’s results by comparing them with the clinicians’ response and the final outcome of the patient.

## 2. Materials and Methods

### 2.1. Study Design and Setting

A retrospective study was conducted at the Hamad Trauma Center (HTC) in Qatar between 1 June 2016, and 30 May 2021, which included all patients diagnosed with TBI, regardless of their age or gender. However, patients who had penetrating injuries, were brought dead to the hospital, or were transferred alive from other hospitals were excluded from the study. The patients were treated as per the guidelines of the Advanced Trauma Life Support (ATLS) protocol, and after stabilization, blood samples were taken for various tests including arterial blood gases, blood chemistry, serum electrolytes, and hematic biometry.

The study utilized data from both the Qatar Trauma Registry (QTR) and electronic medical records (CERNER) to identify cases of TBI based on ICD-10-CM codes. The QTR is a major contributor to the American College of Surgeons Committee on Trauma’s (ACS-COT) National Trauma Databank (NTDB), which is the world’s largest repository of trauma registry data. The registry’s biannual contributions to the ACS-COT’s Trauma Quality Improvement Program (TQIP) include wide-ranging hospital performance reports with comparative standard data that are distributed to participating institutions.

Patient demographics, injury mechanism, head trauma type, injury severity scores, serum electrolyte and biomarker levels at admission, interventions and procedures, complications during hospitalization, and in-hospital mortality were among the data collected. The study was approved by the Institutional Review Board (MRC#01-21-501) of the Hamad Medical Corporation.

### 2.2. Definitions

The Glasgow Coma Scale (GCS) was used to assess the level of consciousness after a head injury, with scores ranging from 3 to 15. A GCS score of less than 8 indicated a severe injury, a score of 9–12 indicated moderate injury, and a score of 13 or higher indicated minor injury [41]. The Abbreviated Injury Scale (AIS) was used to measure the severity of injuries to different body regions, with minor (AIS = 1), moderate (AIS = 2), serious (AIS = 3), severe (AIS = 4), critical (AIS = 5), and non-survivable injuries (AIS = 6) [42]. The AIS scores for the three most severely injured body parts were squared and added to generate the Injury Severity Score (ISS) to measure the overall injury severity for patients with multiple injuries. The range of ISS score was from 0 to 75, where a score of 1 to 8 was considered minor, 9 to 15 was moderate, 16 to 24 was serious, 50 to 74 was critical, and a score of 75 was non-survivable [43].

According to the standards of our institution, the following values are normal levels of serum electrolytes: Sodium of 135 to 145 mEq/L, potassium of 3.6 to 5.2 mmol/L, magnesium of 0.65 to 1.05 mmol/L, calcium of 2.2 to 2.7 mmol/L, and phosphate of 1.1 to 1.45 mmol/L. The outcome of interest in this study was in-hospital mortality which refers to the death of a TBI patient occurring during their hospital stay, regardless of the reason for admission or the duration of their hospitalization.

### 2.3. Feature Selection for ML

We selected relevant features for our model by conducting a chi-square test between patients who survived and died in the hospital and considering both the statistical and clinical relevance of the variables (as shown in Table 1). A total of 13 features were selected for our study’s models, which included GCS, ISS, activated partial thromboplastin time (aPTT), prothrombin time (PT), international normalized ratio (INR), hemoglobin (Hb), lactic acid, initial serum sodium, initial serum potassium, initial serum calcium, initial serum magnesium, initial serum phosphate, and bicarbonate level. The Spearman correlation coefficient between these selected features and the outcome was calculated, and all were found to have an absolute value greater than 0.113.

The comparative analysis between survivors and non-survivors found that age was a statistically significant factor in mortality, but the age difference between the groups was only approximately 5 years and not clinically significant. The ISS and GCS were included in the prediction model to account for the severity of injury while excluding factors such as type of TBI (e.g., subarachnoid hemorrhage, subdural hematoma) and associated injuries (e.g., chest, abdomen, pelvis, spinal injuries). Hemoglobin (Hb) was also included as a feature to represent blood loss and eliminate the need for additional parameters such as blood transfusion.

### 2.4. Imputation for Missing Values

All features used had less than 1% missing values, which were imputed with the median for continuous variables and the mode for categorical features. The data were then randomly split into 80% training and 20% testing datasets.

### 2.5. Prediction Models

The output variable in the study was the mortality of patients. Four different binary classification supervised ML models were utilized for the prediction of mortality. In contrast to unsupervised learning, a supervised ML model is guided by the correct output labels in the training data, where the algorithm attempts to determine patterns in the data without any labeled output information. The models include Logistic Regression (LR), SVM, RF, and XgBoost (Extreme Gradient Boosting). These models were chosen based on their ability to explain the results, the complexity of the model, and the size and type of dataset used.

SVM is an ML algorithm that can be utilized for both linear and non-linear classification tasks [15]. The choice of kernel function is crucial for achieving the optimal separation of classes in SVM. In this case, the Linear kernel was selected because it provided the best results. A kernel function is a mathematical function that transforms input data into a higher-dimensional feature space, where it is easier to separate classes of data. Complex kernel functions, such as polynomial or radial basis function kernels, may be used for data that are not linearly separable.

LR is a widely used method for predicting binary outcomes and can also be used for multiple outcomes [16].

RF is a popular supervised method that employs an ensemble approach by combining multiple decision trees through the bagging technique to increase the accuracy and robustness of the model [17,18].

XGBoost is an ensemble learning algorithm that utilizes a decision tree-based framework and the gradient boosting method. It employs a neural network-inspired approach to structure unstructured data and make predictions [19].

### 2.6. Model Training, Evaluation, and Performance Metrics

The dataset in our study was randomly split into 80% training and 20% testing cohorts in a stratified fashion. Despite having a slightly imbalanced dataset, using the Synthetic Minority Over-sampling Technique (SMOTE) for up-sampling did not lead to improved performance of the models. The data were standardized using sklearn.preprocessing.StandardScaler in Python (version 3.9). The aim of standardizing was to bring the features to a similar scale, which is expected to improve the performance of the models.

SVM and LR are simple models, while RF and XgBoost are ensemble models that use bagging and boosting methods with decision tree base models. Five-fold cross-validation was used with sklearn.model_selection.GridSearchCV for hyperparameter tuning in all models.

In the case of using the Linear SVM, the hyperparameter that was tuned was the regularization parameter, also known as the inverse regularization strength (C). A wide range of logarithmic values was initially considered for C, and then refined by selecting values that yielded the lowest error. The same factor was adjusted for the LR models with L2 regularization. In RF, the following parameters were considered for hyperparameter tuning: Number of trees in the forest (n_estimators), max number of features considered for splitting a node (max_features), max number of levels in each decision tree (max_depth), min number of data points placed in a node before the node is split (min_samples_split), and min number of data points allowed in a leaf node (min_samples_leaf). A similar method to finding the best C in SVM was used to obtain n_ estimators and max_depth. For min_samples_split and min_samples_leaf, the lists in [2,5,10] and [1,2,4] were used. The learning rate (eta), minimum split loss (gamma), maximum depth (max_depth), and colsample_bytree (this is a family of parameters for subsampling of columns) were considered for tuning the XGBoost model. Every model was repeated with different training and testing set samples for the repeatability of the results.

The AUROC curve was plotted to evaluate the models’ performance and check for overfitting. The ensemble models showed some difference in log loss values between the training and testing sets compared to SVM and LR. In addition to AUROC and log loss, a confusion matrix was plotted for a more accurate understanding of the model performance.

Accuracy, specificity, sensitivity, and F1 score were calculated from the matrix. Accuracy measures the overall correctness of the predictions by the model, which is the ratio of the number of correct predictions to the total number of predictions made by the model. Sensitivity (Recall) is the proportion of actual positive cases that are correctly recognized by the model. It is calculated as the ratio of the true positive (TP) predictions to the sum of true positive and false negative (FN) predictions. Specificity calculates the proportion of actual negative cases that are correctly recognized by the model. It is calculated as the ratio of the true negative (TN) predictions to the sum of true negative and false positive (FP) predictions. The F1 score is a harmonic mean of precision and recall (sensitivity). It is a combined metric that considers both precision and recall, and it balances the trade-off between them. F1 score is measured as 2 × (precision × recall)/(precision + recall). It is a good metric to use when the classes are imbalanced, i.e., when there is a significant difference in the number of samples in each class.

Finally, feature importance was plotted, which is an index that shows how important each feature is in determining the outcome. Decision trees and RF have built-in feature importance measures based on the reduction in impurity achieved by splitting on a particular feature. Other models, such as linear regression and logistic regression, use coefficients to represent the importance of each feature. The feature importance scores are often used to rank the input features in terms of their importance so that the most important features can be selected for further analysis or used to develop simplified models with fewer input features.

Logarithmic Loss, also known as Log Loss or Cross-Entropy Loss, is a commonly used loss function in ML for classification problems. It evaluates the performance of a classification model where the output is a probability value from 0 to 1. The lower the Log Loss value, the better the performance of the classification model.

The formula for Log Loss is:Log Loss = −1/N × Σ (y × log(y_hat) + (1 − y) × log(1 − y_hat))
where N is the number of instances, y is the true label (0 or 1), and y_hat is the predicted probability of the model for the given instance.

The percentage difference in log loss between the testing and training sets can be used as a measure to evaluate the generalization performance of a classification model.

The formula for calculating the percentage difference in log loss between the testing and training sets for a given model is:% difference in log loss = (log loss (test set) − log loss (train set))/log loss (train set) × 100
where log loss (testing set) and log loss (training set) are the log loss values of the testing and training sets, respectively, for the given model.

The percentage difference in log loss indicates how much the log loss of the testing set differs from the log loss of the training set, expressed as a percentage of the log loss of the training set. A positive percentage difference shows that the model is overfitted, as it performs better on the training set than on the testing set. A negative value of % difference in log loss indicates an error in our model as it indicates that the model performs better for the unseen testing set than the training set, which is very unlikely.

Figure 1 provides an overview of the data analysis process used in this study, which includes a flowchart of the key steps involved in understanding the objectives and constraints, data collection, investigative data analysis, data groundwork, model selection, evaluation, and visualization. The process involved collecting data from the Hamad Trauma center and conducting a statistical and graphical exploration of the raw data, cleaning, imputation, featurization, and partitioning into training and testing sets. The figure also shows the development and testing of an interpretable, supervised binary classification model that was evaluated and visualized using graphical and tabular formats.

## 3. Results

### 3.1. Characteristics of Study Population

This 5-year study included 922 patients hospitalized with TBI. The majority of patients were male (94%), with a mean age of 32 years. The leading cause of injury was road traffic accidents (59%). The most prevalent type of TBI among the study population was subarachnoid hemorrhage (SAH, 42%) followed by subdural hematoma (SDH, 35%). The majority of the patients (74%) had a severe GCS (3–8). The mean AIS for the chest, abdomen, cervical, thoracic, and lumbar spine, and pelvis region were greater than two. The mean values of initial sodium, potassium, and magnesium levels were within their normal ranges, but the mean values of initial calcium and phosphates were lower than the normal lower limits. Intubation was performed in 90% (*n* = 827) of the patients, the Massive Transfusion Protocol (MTP) was implemented in 15% (*n* = 138), and craniotomy or craniectomy was performed in 21% (*n* = 192) of the patients. Ventilator-associated pneumonia (VAP) was reported in 121 (13%) patients. The median length of stay on a ventilator, in the ICU, and in the hospital were 5 days, 9 days, and 17 days, respectively. The in-hospital mortality rate was 22% (*n* = 204) (Table 1).

### 3.2. Comparing Machine Learning Model Performance

When comparing the performance of the different models, the primary metric used was the Receiver Operating Characteristic Area Under the Curve (AUROC) value. This value is depicted in Figure 2 and is a commonly used metric for evaluating the performance of binary classifiers. However, due to the imbalanced nature of the data, accuracy alone was not considered a performance measure. Instead, the sensitivity of the model was also considered to be a crucial metric, as it reflects the ability of the model to accurately identify positive cases.

The AUROC scores for the SVM, LR, XgBoost, and RF models were 0.86, 0.84, 0.85, and 0.86, respectively. Despite the good AUC scores for the XgBoost and RF models, they displayed a significant difference in Log loss between the training and testing sets, with a % difference in log loss of 79.5 and 41.8, respectively. This indicates that these models are overfitted compared to the other models, meaning they are performing well on the training data but not as well on new unseen data.

To gain a deeper understanding of the dependence of each feature on the prediction model, feature importance was calculated and plotted in Figure 3, Figure 4, Figure 5 and Figure 6. These plots show the relative importance of each feature in the prediction model and reveal that in cases where all the data are not available or where features need to be reduced, the most important features for predicting mortality were aPTT, INR, lactic acid, PT, magnesium, ISS, and GCS.

## 4. Discussion

### 4.1. Evaluation of four Models and Feature Selection Considerations

The aim of the study was to evaluate ML methods to predict the mortality of TBI patients admitted to the ICU over a 5-year period (*n* = 922). The in-hospital mortality was 22% (*n* = 204). Four ML models including SVM, LR, XgBoost, and RF were evaluated, and all four models demonstrated a predictive accuracy of over 80% for mortality. The SVM model showed the best overall performance. The trend in feature importance across all models indicated that the most crucial factors in prediction were aPTT, INR, ISS, PT, and lactic acid. Magnesium was also found to be a significant predictor of mortality among serum electrolytes.

The factors that determine the amount of data needed to construct an ML model include the complexity of the model, the number of features, the presence of noise in the data, and the expected performance of the model. While it is suggested to have a minimum of several thousand data points for a robust ML model, having 922 data points for TBI patients can suffice, though accuracy may not be optimal. Nevertheless, we employed a dataset that is diverse, representative of the population, and free from bias, avoiding overfitting, a scenario where a model performs optimally on the training data but poorly on new and unseen data.

A critical component of creating an ML model for TBI prediction is feature selection. By removing features that are unnecessary or redundant, feature selection aims to choose a subset of features that are most pertinent to the current issue. Reducing overfitting, enhancing interpretability, and accelerating the training process aids in improving the model’s performance. Previous studies indicate that several ML algorithms can be employed for predicting mortality in TBI patients.

The study by Amorim et al. [10] found that high TBI severity was linked with high rates of epidural, subarachnoid, subdural, and intracerebral hemorrhage levels. Additionally, they also found worse GCS scores associated with high TBI severity scores. During the feature selection process, we omitted the rates of hemorrhage levels as they were already reflected in the TBI severity scores. Coagulation features such as blood transfusion in our study can play an important role in predicting TBI severity using ML models. However, Hb levels in our study can potentially be a useful feature in predicting mortality instead of blood transfusion since it was more easily accessible at the time of admission.

### 4.2. Comparing Different ML Algorithms

In a study led by Tu et al. [19], the aim was to predict the in-hospital mortality risk for 18,249 adult TBI patients using 12 feature variables. The variables were selected based on criteria such as availability, significance, and interpretability. The predictive model was developed using six ML algorithms such as LR, RF, SVM, LightGBM (Light Gradient Boosting Machine), XGBoost, and Multilayer Perceptron (MLP). The results demonstrated that all these models performed well, with the AUC ranging from 0.851 to 0.925. The optimal model was selected through a grid search with 5-fold cross-validation. The results were based on a classification threshold of 0.5, where a result equal to or greater than the threshold indicated a positive outcome (mortality), and a result less than the threshold indicated a negative outcome (survival). The LR-based model was found to have the highest AUC of 0.925, making it the best model for predicting mortality in TBI patients.

Gradient-based One-Side Sampling (GOSS), a cutting-edge method used by LightGBM, speeds up training and requires less memory. When choosing a subset of the data for training, GOSS keeps the data points that are the most informative while discarding the ones that are less helpful. Due to less overfitting, the algorithm may focus on the most pertinent facts. The MLP machine learning algorithm is strong and adaptable, and it may be used for a variety of tasks. To obtain good performance, it might be sensitive to the selection of hyperparameters and may require a significant amount of training data and computational power. Additionally, MLP is susceptible to overfitting, which can be prevented by employing strategies such as early halting and regularization.

Similar to the study by Tu et al. [19], we evaluated four ML algorithms (SVM, LR, RF, and XgBoost) using a combination of clinical relevance measures and a chi-square test to determine the most important features for predicting mortality. The final list of 13 selected features included GCS, ISS, APTT, PT, INR, Hb, Lactic Acid Level, Bicarbonate Level, and Initial Serum levels of Sodium, Potassium, Calcium, Magnesium, and Phosphate. This approach was different from that used in the study by Tu et al. [16], where they used a correlation analysis to identify the leading feature variables for predicting mortality, including the GCS, pupillary light reflex scores, the Taiwan Triage and Acuity Scale (TTAS), pupil size, and patient age. The analysis showed that the GCS, both pupillary light reflex scores, and TTAS had a negative correlation with mortality, while age and pupil size had a positive correlation with mortality during hospitalization. The TTAS is a system used in EDs in Taiwan to prioritize patients based on the severity of their condition [17]. It considers various factors, including vital signs and clinical indicators, to determine the patient’s level of acuity. TTAS has been shown to be reliable and accurate in identifying patients who need urgent care, making it effective in reducing waiting times and improving care in emergency departments.

The study by Warman et al. [18] aimed to evaluate the accuracy of ML models in predicting in-hospital death rates in high-income and low- and middle-income countries. The researchers used data from two registries, NTDB and Mulago National Referral Hospital (MNRH), and six ML algorithms (LR, RF, SVM, K-nearest neighbors (K-NN), Gaussian naive Bayes, and XGBoost). The best ML model was chosen through cross-validation. The XGBoost algorithm showed the best performance in high-income countries with an AUROC of 0.91 and an AUPRC of 0.53. In low- and middle-income countries, the best algorithm was the SVC model with an AUROC of 0.89 and an AUPRC of 0.54. The models were well-calibrated and had similar performance for AUROC and Area Under the Precision-Recall Curve (AUPRC) optimization.

The study by Rau et al. [22] aimed to develop an ML model for predicting mortality in patients with moderate to severe TBI. The researchers enrolled hospitalized adult patients from the Trauma Registry System between 2009 and 2015, and only included patients with head injuries with an AIS score of 3 or higher. The study used demographic and injury characteristics, as well as laboratory data, to predict patient mortality using LR, SVM, decision trees, naive Bayes, and artificial neural networks (ANNs). The performance of the ML models was evaluated using accuracy, sensitivity, specificity, and the AUC. In the training set, all five models had a specificity greater than 90%, with the artificial neural network having the highest sensitivity (81%) and the highest AUC (0.968). In the testing set, the ANN had the highest sensitivity (84%) in predicting mortality.

Abujaber et al. [9] conducted a study aimed to develop an ML model to predict in-hospital mortality in patients with moderate to severe TBI who were on mechanical ventilation. The study included 785 adult patients with TBI hospitalized in our trauma center. The CT findings and demographic characteristics were used as predictors, and two models, ANN and LR, predicted in-hospital mortality. The models accomplished good performance as the accuracy was over 80% and AUROC over 87%. Albeit LR showed high overall performance when compared to ANN with accuracy and AUROC of 87% and 90.5%, respectively. Diagnosis of TBI, TBI severity, blood transfusion requirement, heart rate on ED admission, and age of the patients were important predictors of in-hospital mortality for TBI patients on mechanical ventilation.

Bruschetta et al. [20] conducted a study aimed to compare classical linear regression models with ML algorithms for predicting the prognosis of TBI patients. The study evaluated TBI patients at baseline, 3 months after the event, and at discharge. The study compared the performance of a classical linear regression model with SVM, KNN, naive Bayes, and decision tree algorithms, as well as an ensemble ML approach. The study found that the accuracy was similar among the linear regression and ML algorithms when using a two-class approach. The naive Bayes algorithm had the worst performance. The study highlights the importance of comparing traditional regression models with ML algorithms, particularly when using a small number of predictor variables.

### 4.3. Translating the Model into Clinical Practice and Validation Steps

In clinical practice, the data obtained during the initial hours of admission can be inputted into the trained ML model to generate a predictive response. This outcome can serve as an alert system for clinicians, providing valuable insights into the patient’s condition and facilitating timely intervention.

To ensure the reliability and clinical applicability of our ML model, we propose a comprehensive validation process comprising multiple steps. Firstly, we compared the model’s predictions with the clinicians’ response in real-time, enabling an assessment of the model’s accuracy and its ability to align with expert judgment.

In addition to retrospective analysis, a prospective study can be conducted, where the model’s predictions will be prospectively collected and compared with the subsequent patient outcomes. This longitudinal evaluation will provide further evidence of the model’s performance and generalizability, as well as its ability to aid clinical decision-making in real-world settings.

By incorporating these validation steps, we aim to address the importance of translating the ML model into clinical practice and ensuring its robustness. The comparison with clinicians’ responses and patient outcomes will provide valuable insights into the model’s performance, enhance its credibility, and enable confident decision-making based on its predictions.

The proposed validation process not only validates the accuracy of the model but also assesses its clinical utility and potential impact on patient care. These steps are crucial in establishing trust in the model’s predictions and fostering its successful integration into routine clinical practice.

### 4.4. Limitations

In this study, limitations such as the small sample size, which complicated data processing and model training, were encountered. However, the dataset and variables used were still comparable to previous studies and were obtained from a validated trauma registry. Other limitations include possible miscoding or bias in feature variables due to unrecognized confounders, as well as the lack of evaluation of other variables that could influence TBI patient outcomes. The study’s results may not be generalizable to other hospitals, and further external validation is needed for more diverse samples.

## 5. Conclusions

Multiple ML algorithms can predict mortality in TBI patients, but some may face issues due to sensitivity to hyperparameters and overfitting. Feature selection is critical in creating an ML model for TBI prediction. Future studies should aim to collect a more diverse, representative, and unbiased dataset to develop robust ML models for TBI prediction.

## Figures and Tables

**Figure 1 diagnostics-13-02605-f001:**
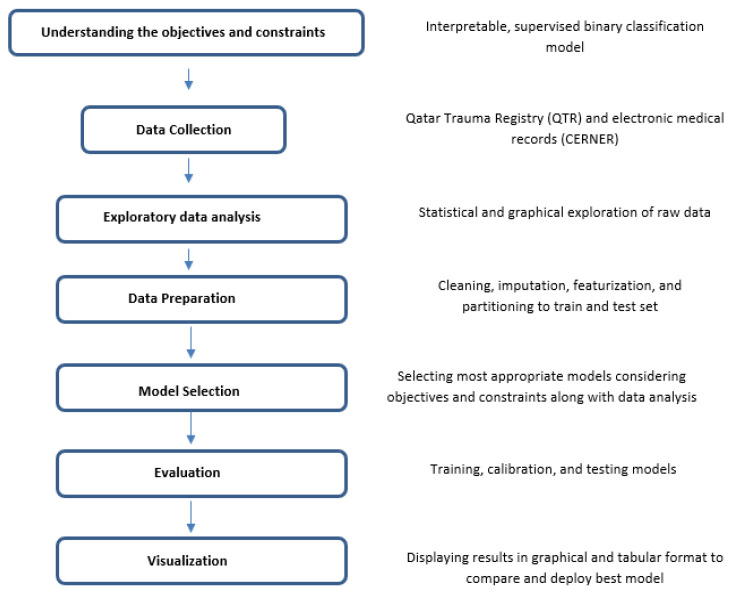
An overview of the key steps involved in machine learning approach.

**Figure 2 diagnostics-13-02605-f002:**
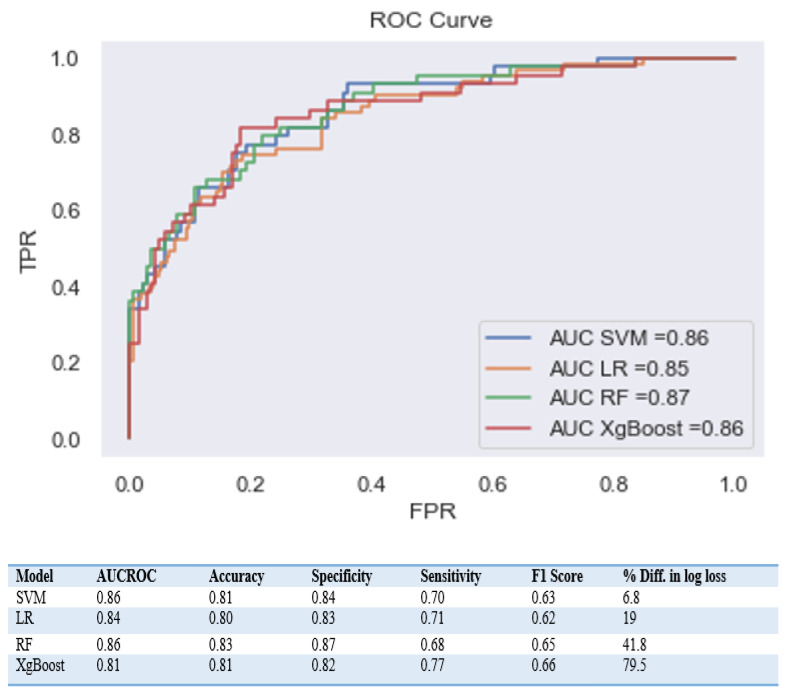
Comparison of performance metrics for different machine learning approaches.

**Figure 3 diagnostics-13-02605-f003:**
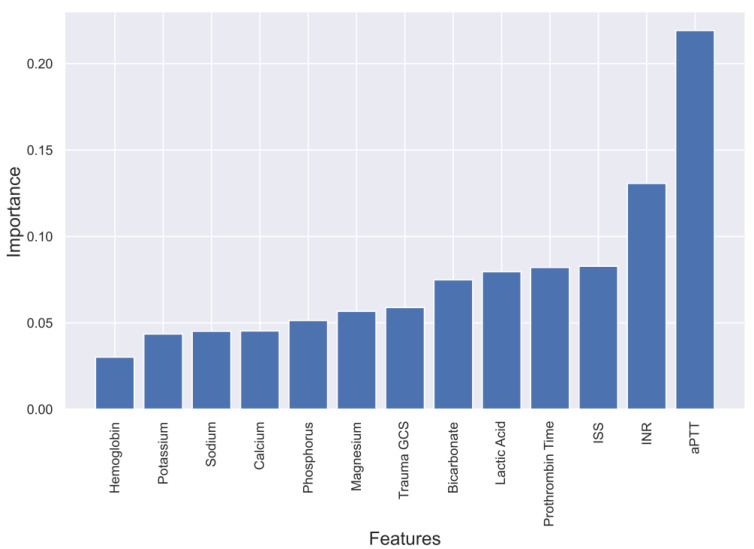
Feature importance in XgBoost model predicting mortality of TBI patients.

**Figure 4 diagnostics-13-02605-f004:**
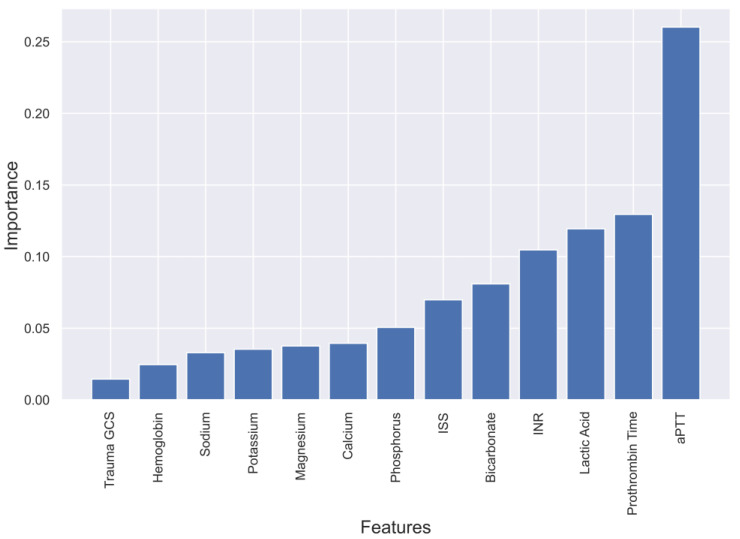
Feature importance in Random Forest model predicting mortality of TBI patients.

**Figure 5 diagnostics-13-02605-f005:**
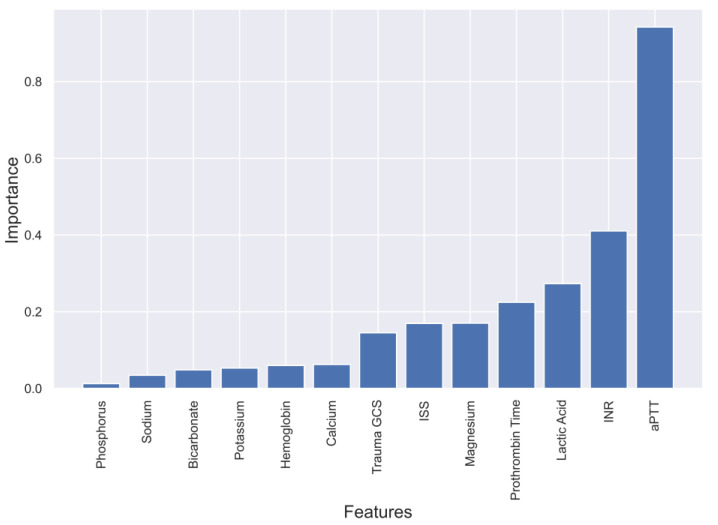
Feature importance in Linear SVM model predicting mortality of TBI patients.

**Figure 6 diagnostics-13-02605-f006:**
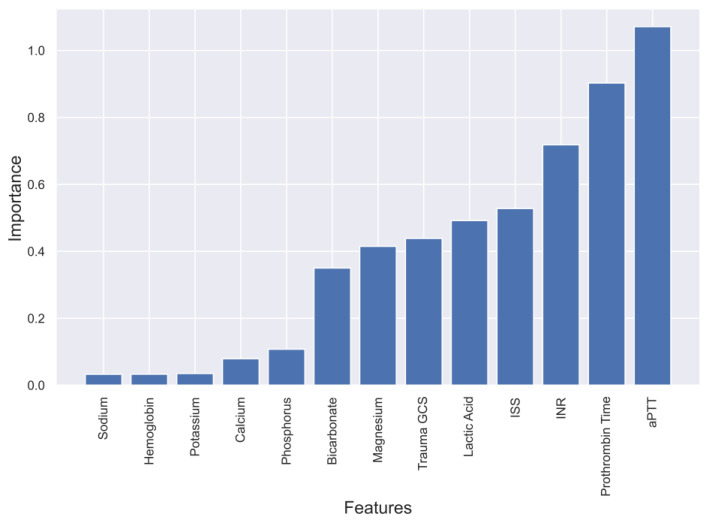
Feature importance in Logistic Regression model predicting mortality of TBI patients.

**Table 1 diagnostics-13-02605-t001:** Bio-clinical characteristics and outcomes for hospitalized Traumatic Brain Injury (TBI) patients (*n* = 922).

Variable	Value
Age (mean ± standard deviation)	32.2 y ± 15.0
Males	863 (93.6%)
Mechanism of injury (*n*, %)	
• Falls	230 (24.9%)
• Road traffic injuries	545 (59.1%)
• Other	147 (15.9%)
Types of head injury (*n*, %)	
• Epidural hematoma	204 (22.1%)
• Subdural hematoma	321 (34.8%)
• Subarachnoid hemorrhage	387 (42.0%)
• Compression of basal cisterns	110 (11.9%)
• Effacement of Sulci	171 (18.5%)
• Midline Shifts	206 (22.3%)
Glasgow Coma Scale (GCS) classification (*n*, %)	
• Mild (GCS 14–15)	66 (7.2%)
• Moderate (GCS 9–13)	166 (18.0%)
• Severe (GCS 3–8)	681 (73.9%)
• Injury Severity Score (ISS) (median, IQR)	27 (18–34)
Initial Serum Electrolyte Levels [median, interquartile range (IQR)]	
• Initial serum sodium	141.0 (139–143)
• Initial serum potassium	3.8 (3.4–4.1)
• Initial serum calcium	2.0 (1.8–2.1)
• Initial serum magnesium	0.7 (0.6–0.8)
• Initial serum phosphate	0.9 (0.7–1.2)
Other Clinical Parameters (median, IQR)	
• Bicarbonate level	19.6 (16.7–23.0)
• Lactic acid level	2.9 (2.0–4.3)
• Prothrombin time	12.0 (11.1–13.5)
• Activated partial thromboplastin time	26.2 (24.0–31.0)
• International normalized ratio	1.1 (1.1–1.3)
• Hemoglobin	13.0 (11.3–14.4)
• Glucose	8.0 (6.7–10.1)
Interventions and Procedures (*n*, %)	
• Intubation	827 (89.7%)
• Blood transfusion	480 (52.1%)
• Massive transfusion protocol activation	138 (15.0%)
• Intracranial pressure monitoring	208 (24.6%)
• Craniotomy/craniectomy	192 (20.8%)
Length of stay in days (LOS) (median, IQR)	
• Mechanical ventilator	5 (2–11)
• Intensive care unit	9 (4–17)
• Hospital	17 (7–32)
In-hospital mortality	204 (22.1%)

## Data Availability

All data generated or analyzed during this study are included in this article. Data are accessible upon agreement with the Medical Research Centre, Qatar.

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
