# Peer review of "Machine Learning Approach for the Prediction of In-Hospital Mortality in Traumatic Brain Injury Using Bio-Clinical Markers at Presentation to the Emergency Department"

_diagnostics, 2023, doi:10.3390/diagnostics13152605_

Round 1
Reviewer 1 Report
In this work, the authors attempted to identify predictors of in-hospital mortality in TBI patients 15 using ML algorithms. The results seem reasonable for a decent model prediction. The introduction section is too short to justify the rationale for doing this work. There are several works done previously, and the authors have not identified the gaps or limitations of these methods, and how they are improving them. I suggest, more thorough literature review to strengthen the need to do this work. This study seems to be a repeat to validate other work, without new information provided. What is the novelty of this work?
Also, please provide a discussion on how this model will be translated into clinical practice. What are the validation steps?
Author Response
Q: The results seem reasonable for a decent model prediction. The introduction section is too short to justify the rationale for doing this work. There are several works done previously, and the authors have not identified the gaps or limitations of these methods, and how they are improving them. I suggest, more thorough literature review to strengthen the need to do this work.
Response: an extensive literature review was performed which provided a good understanding of what is known and what is done previously, and the gap identified was written in the introduction as "Given that the use of ML techniques in trauma is still in the emerging phase, it remains inconclusive in the literature regarding the most appropriate method to be used and which performance indicators are most suitable for validation". Regarding the novelty, it is also written as " Being a developed country in the Arabian Gulf region with a small geographical size and population, limited research has been conducted in this specific region regarding the utilization of machine learning techniques, particularly in the context of TBI patients. By conducting this study in Qatar, valuable insights can be gained that are specific to the region, contributing to the existing body of knowledge in the field of ML applied to TBI". The majority of TBI victims in Qatar is young males and the utility of ML for early triage and prediction would be useful to save more lives.
Q: This study seems to be a repeat to validate other work, without new information provided. What is the novelty of this work? Also, please provide a discussion on how this model will be translated into clinical practice. What are the validation steps?
Reply: The novelty of the work has been discussed above. To address the clinical practicing and validation part, a separate portion is added in the discussion section. The portion discusses that the data obtained during the initial hours of admission can be in-putted into the trained ML model to generate a predictive response. This outcome can then be used to alert clinicians about the patient's condition and validate the model's results by comparing them with the clinicians' response and the final outcome of the patient.
In the revised manuscript , we added new portion in the discussion (4.3 Translating the Model into Clinical Practice and Validation Steps) .
Reviewer 2 Report
The authors present a study of the effectiveness of four machine learning algorithms to predict mortality in TBI patients.
Scientific matters to be addressed:
paragraph starting on line 192: The discussion of the relationship of the percent change in log loss to overfitting/underfitting is not correct. While I agree that high positive value would suggest overfitting. A consistent negative value is unlikely. That would mean that the model consistently does better on test data than training data. If that happened, I would be looking for an error in my training process. At worst the percent difference log loss should be zero (on average) - meaning learning is consistent (not necessarily good or bad) across both training and test. A model that randomly guesses is clearly underfit and should (on average) have the same accuracy across training and test. This discussion needs to be corrected.
There is also not enough detail about model training. For example, RFs are typically somewhat resistant to overfitting. What hyperparameters were adjusted to try to correct for overfitting and how? I see no evidence that a serious attempt was made to address the overfitting in either ensemble of trees, and I suspect that if there had been one, RFs would not overfit.
Also, with such a small data set, I believe the experiment would be particularly sensitive to an unlucky random train/test split. Were the reported result computed by averaging multiple runs? DId you consider a stratified partitioning of the data?
Suggested edits:
line 37: "highest"? Compared to what?
line 338: This paragraph needs to be deleted - it is a reminder to the authors.
Capitalize the title in the first reference. Other comments are noted above.
Author Response
Q: paragraph starting on line 192: The discussion of the relationship of the percent change in log loss to overfitting/underfitting is not correct. While I agree that high positive value would suggest overfitting. A consistent negative value is unlikely. That would mean that the model consistently does better on test data than on training data. If that happened, I would be looking for an error in my training process. At worst the percent difference log loss should be zero (on average) - meaning learning is consistent (not necessarily good or bad) across both training and test. A model that randomly guesses is clearly underfit and should (on average) have the same accuracy across training and test. This discussion needs to be corrected.
Reply: I have reviewed and updated the paragraph to reflect the corrected explanation. The revised statement now clarifies that a positive percentage difference in log loss indicates overfitting, as the model performs better on the training set than on the test set. Conversely, a negative value of the percentage difference suggests an error in the model, as it is unlikely for the model to consistently perform better on the unseen test set than on the training set. I appreciate your valuable input, and I have incorporated the necessary changes to ensure the accuracy of the discussion
Q: There is also not enough detail about model training. For example, RFs are typically somewhat resistant to overfitting. What hyperparameters were adjusted to try to correct for overfitting and how? I see no evidence that a serious attempt was made to address the overfitting in either ensemble of trees, and I suspect that if there had been one, RFs would not overfit.
Reply: I have made the necessary additions to address this concern. In the case of Linear SVM, the regularization parameter (C) was tuned by considering a wide range of logarithmic values and selecting the ones that minimized the error.
For the Logistic Regression (LR) models with L2 regularization, a similar approach was followed, adjusting the regularization factor. In the Random Forest (RF) models, several hyperparameters were considered for tuning, including the number of trees in the foreset (n_estimators), the maximum number of features considered for splitting a node (max_features), the maximum number of levels in each decision tree (max_depth), the minimum number of data points required for splitting a node (min_samples_split), and the minimum number of data points allowed in a leaf node (min_samples_leaf). The best values for n_estimators and max_depth were obtained using a similar method to the one used for finding the best C in SVM. Additionally, a list of values (e.g., [2, 5, 10]) was explored for min_samples_split and min_samples_leaf.
In the XGBoost model, hyperparameters such as the learning rate (eta), minimum split loss (gamma), maximum depth (max_depth), and colsample_bytree (a parameter for subsampling columns) were considered and tuned.
To ensure the repeatability of the results, every model was repeated with different train and test set samples.
Q: Also, with such a small data set, I believe the experiment would be particularly sensitive to an unlucky random train/test split. Were the reported result computed by averaging multiple runs? Did you consider a stratified partitioning of the data?
Reply: I have addressed this issue in the paper by incorporating your suggestions. The dataset in our study was randomly split into an 80% training and 20% testing cohort, using a stratified approach to ensure a representative distribution of classes. Additionally, to enhance the reliability of the results, every model was repeated with different train and test set samples, allowing for repeatability and mitigating the impact of an unlucky random split.
Suggested edits:
line 37: "highest"? Compared to what?
Response: The sentence actually started "among all forms of injuries,.." Therefore it is obvious that compared to other injuries. However, "highest" is changed to "high" to avoid confusion
line 338: This paragraph needs to be deleted - it is a reminder to the authors.
Response: deleted, thanks
Round 2
Reviewer 1 Report
The authors have made reasonable edits based on the reviewer's comments.
I recommend publication of this manuscript.
Reviewer 2 Report
I believe that the authors adequately address my earlier concerns.